# Relationship between glass transition temperature, and desiccation and heat tolerance in *Salmonella enterica*

**Kyeongmin Lee**[1☯], **Masaki Shoda**[1☯], **Kiyoshi Kawai**[2], **Shigenobu Koseki**[1]*

**1** Graduate School of Agricultural Science, Hokkaido University, Sapporo, Japan, **2** Graduate School of Integrated Sciences for Life, Hiroshima University, Higashi-Hiroshima, Hiroshima, Japan

☯ These authors contributed equally to this work.
* koseki@bpe.agr.hokudai.ac.jp

**Data Availability Statement:** All relevant data are within the manuscript.

**Funding:** This work was supported by the Japan Society for the Promotion of Science (JSPS) KAKENHI (Grant JP 18H02148) granted to SK. The

## Abstract

Pathogenic bacteria such as *Salmonella enterica* exhibit high desiccation tolerance, enabling long-term survival in low water activity ($a_w$) environments. Although there are many reports on the effects of low $a_w$ on bacterial survival, the mechanism by which bacteria acquire desiccation tolerance and resistance to heat inactivation in low-$a_w$ foods remains unclear. We focused on the glass transition phenomenon, as bacteria may acquire environmental tolerance by state change due to glass transition. In this study, we determined the glass transition temperature ($T_g$) in *S. enterica* serovars under different $a_w$ conditions using thermal rheological analysis (TRA). The softening behaviour associated with the state change of bacterial cells was confirmed by TRA, and $T_g$ was determined from the softening behaviour. $T_g$ increased as the $a_w$ decreased in all *S. enterica* serovars. For example, while the $T_g$ of five *S. enterica* serovars was determined as 35.16°C to 57.46°C at 0.87 $a_w$, the $T_g$ of all the five serovars increased by 77.10°C to 83.30°C at 0.43 $a_w$. Furthermore, to verify the thermal tolerance of bacterial cells, a thermal inactivation assay was conducted at 60°C for 10 min under each $a_w$ condition. A higher survival ratio was observed as $a_w$ decreased; this represented an increase in $T_g$ for *Salmonella* strains. These results suggest that the glass transition phenomenon of bacterial cells would associate with environmental tolerance.

## Introduction

Outbreaks of foodborne illnesses caused by dry foods, such as nuts [1–3], chocolate [4–8], cereals [9], and other foods are continuing worldwide [10–12]. Such foods have low water activity ($a_w$) and the growth of bacteria causing foodborne illness is not observed. Therefore, microbiological hygiene control has not been regarded as important. However, cases of foodborne illness caused by dry food occur frequently, which means that pathogenic bacteria continue to survive even in a low-$a_w$ environment. Especially, various *Salmonella* serotypes were found in dry foods associated with outbreaks [1–12]. Indeed, several reports have shown that

funders had no role in study design, data collection and analysis, decision to publish, or preparation of the manuscript.

**Competing interests:** The authors declare no competing interests.

bacteria causing food poisoning such as enterohemorrhagic *Escherichia coli* (EHEC) and *Salmonella*, continue to survive for long periods in a low-$a_w$ environment [13–21]. On the contrary, the rate of bacterial survival decreases under high $a_w$ environments [16,22–24], and it is inferred that $a_w$ and the water content of bacteria have some influence on survival. From these previous studies, we consider that there are some associations between $a_w$ and desiccation tolerance of bacterial cells. However, the mechanism of desiccation tolerance of bacterial cells under low-$a_w$ conditions has not yet been clarified and elucidation of the cause is required.

There is a clue to elucidation of the mechanism of desiccation tolerance in other organisms. For example, extreme environment microorganisms, such as tardigrades and sleeping chironomids that utilize cryptobiosis, are resistant to various external environments such as high temperature, high pressure, as well as dry environments [25–27]. In this study, we assume that bacterial cells would vitrify as well as extreme environmental organisms are considered to acquire environmental stress tolerance. Vitrification of bacterial cells by the glass transition phenomenon might be one of the long-term survival factors in a low-$a_w$ environment. The glass transition phenomenon refers to a state change caused by the increase or decrease in molecular movement in a substance as the temperature and moisture content change [28–30]. The state in which molecular movement is limited, due to the decrease in temperature and $a_w$, is called a glass state and the substance shows physical properties similar to a solid. Since molecular motion is almost stopped in the glass state, the substance or organism shows high tolerance to various environmental stresses such as heat, desiccation, and pressure. In the present study, we hypothesized that bacterial cells are vitrified in low-$a_w$ environments based on the physicochemical properties of solid particles. In other words, we assumed that bacterial cells enter a glass state due to a decrease in molecular movement accompanying a decrease in $a_w$, making it difficult for the bacteria to be influenced by external factors. This, in turn, allows for long-term survival, even in a dry environment.

While aiming to elucidate the survival mechanism of pathogenic bacteria, several studies have reported a decline in the effect of bacterial thermal inactivation in dry foods [31–37], and it is speculated that bacteria in dry foods may exhibit heat tolerance. We believe that the glass transition of bacteria is greatly involved in heat tolerance development as is seen in other organisms living in extreme environments such as tardigrada and sleeping chironomids [25–27].

We hypothesized that there would be a difference among $a_w$ conditions in glass transition temperature ($T_g$) for bacterial cells. Differential scanning calorimetry (DSC) is widely used as a method for measuring $T_g$ [29,38,39]. However, it is difficult to measure $T_g$ of a composite using DSC because the thermogram shows intricate thermal responses [29]. Therefore, here, thermal rheological analysis (TRA) was used to measure $T_g$. TRA, which measures $T_g$ by attaching a temperature control device to a rheometer, is based on the principle of thermal mechanical analysis [28–30]. Previous studies used by TRA investigated the effect of water content on the $T_g$ of cookies [29, 40], hazelnuts [41], and deep-fried food [28]. To conduct the measurements, a sample is compressed at a temperature below $T_g$, and heated above $T_g$ with compression. Then, the $T_g$ of the sample can be determined as a force drop induced by the glass transition. This is a useful method to apply to amorphous powders. By determining $T_g$ values, we could confirm the glass transition of bacterial cells. In addition, we sought to elucidate the influence of $a_w$ on bacterial survival and its relationship with $T_g$. Finally, we aimed to resolve the relationship between the state change of several *Salmonella* serotypes that is known to be present in low water activity foods due to glass transition and the changes in thermal resistance in a desiccation environment. The results obtained here will help to understand bacterial survival in a dry environment, which has not been clarified.

## Material and methods

### Bacterial strains and culturing

*Salmonella enterica* Typhimurium (RMID 1985009 from the Research Institute for Microbial Diseases of Osaka University; isolated from patients in sporadic case), *S. enterica* Chester, *S. enterica* Oranienburg (from the Aomori Prefectural Research Laboratory of Public Health; isolated from dried squid chips associated with an outbreak in 1999), *S. enterica* Stanley (RIMD 1981001 from the Research Institute for Microbial Diseases of Osaka University; isolated from patients in sporadic case), and *S. enterica* Enteritidis (RIMD 1933001 from the Research Institute for Microbial Diseases of Osaka University; isolated from patients in sporadic case) were used in this study.

These serovars were maintained at -80°C in tryptic soy broth (TSB, Merck, Darmstadt, Germany) containing 10% glycerol. The strains were activated after incubating at 37°C for 24 h on tryptic soy ager (TSA, Merck) plates. An isolated colony of each bacterium was then transferred to 5 mL of TSB in a sterile centrifuge tube, incubated at 37°C for 24 h, and then a 100 μL aliquot of cultured bacteria was added to 400 mL TSB and incubated at 37°C for 48 h. The cultured cells were collected by centrifugation (3,000 × $g$, 10 min) and the pellets were resuspended in 5 mL of pure water. Bacterial-cell pellets were obtained by pipetting off the excess water and collected on a plastic plate. The plates were frozen at -80°C for 24 h before drying for 24 h using a freeze dryer (FDU-2200, EYELA, Tokyo, Japan). Dried bacterial cells were crushed, placed in an air-tight container at the desired relative humidity (% RH), which was produced using saturated salt aqueous solutions (43% RH: potassium carbonate, 57% RH: sodium bromide, 75% RH: sodium chloride, and 87% RH: potassium chloride), and stored at 4°C for 48 h. The water activity and temperature in the air-tight container were continuously checked using thermo recorder (TR-72wf, T and D, Nagano, Japan). And the water activity of the bacteria was confirmed by a water activity meter (Aqualab 4TE, Decagon Devices, Washington, USA).

### Determination of glass transition temperature ($T_g$)

Thermal rheological analysis (TRA) was used to measure $T_g$ by attaching a temperature control device to a rheometer (EZ-SX, SHIMADZU, Kyoto, Japan) (illustrated in Fig 1); the analysis is based on the principle of thermal mechanical analysis [28–30]. A dried bacterial cell sample (ca. 100 mg) was placed in the forming die ($\varphi$ = 3 mm) and compacted with a rheometer at ca. 10 MPa. Subsequently, the sample was compressed at ca. 5 MPa ca. for 1 to 3 min and then heated at a rate of approximately 3°C/min until the temperature reached 120°C. Pressure-time data were collected with software attached to the rheometer. In parallel, a thermocouple was attached to the bottom of the forming die and time-temperature data were collected every second using a data logger. Accordingly, the relationship between pressure and temperature data during heating was determined. Since pressure reduction begins at the point at which the bottom temperature of the sample reaches the mechanical $T_g$, the onset temperature of pressure reduction could be regarded as the $T_g$ of the sample [28].

### Thermal inactivation under each water activity ($a_w$) condition

Dried bacterial cells (ca. 50 mg), adjusted to each $a_w$ condition, were placed into a small plastic bag (20 mm x 20 mm), making a thin layer, and vacuum-sealed before submerging in a hot water bath at 60°C for 10 min. In the same manner, dried bacterial cells (ca. 50 mg) were taken before heating and determined the viable cell number as an initial condition. Following heating, the samples were combined with 500 μL of 0.1% peptone water, serially diluted in 0.1%

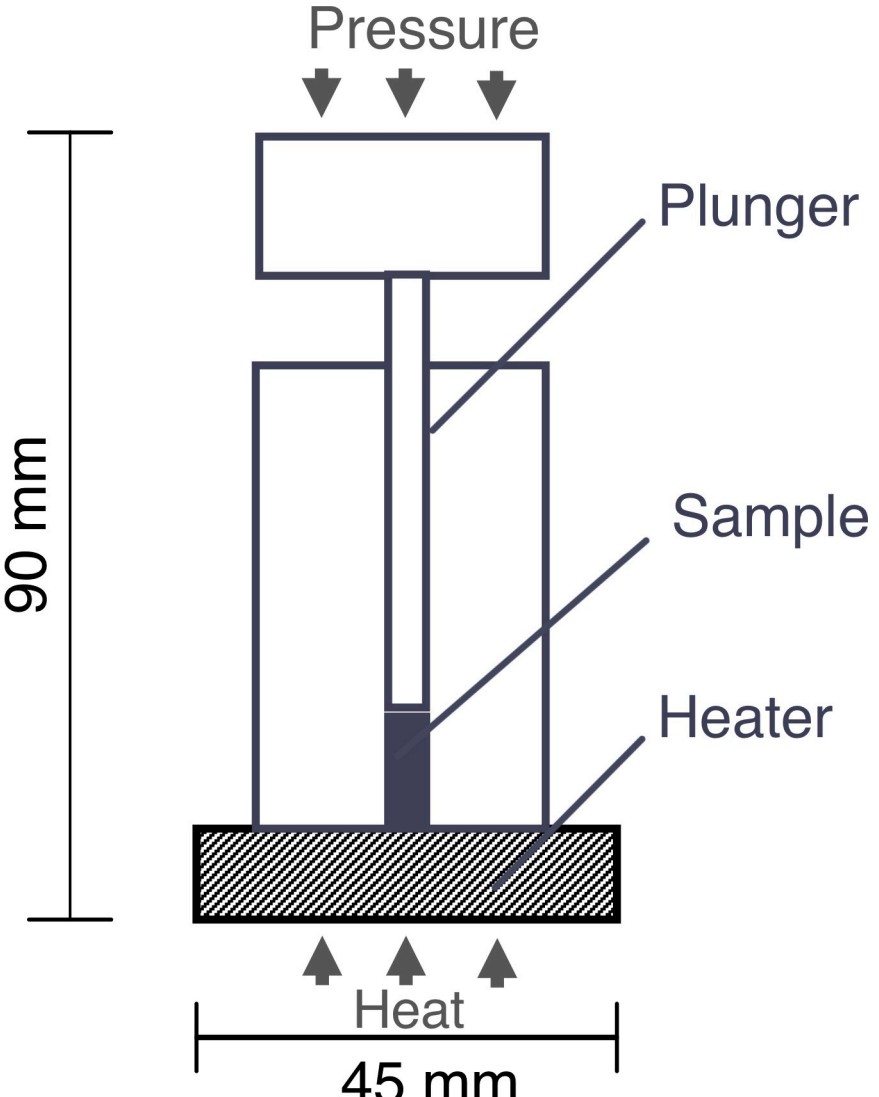

**Fig 1. Schematic drawing of Thermal Rheological Analysis (TRA).**

peptone water, and surface-plated on TSA. The surviving populations were determined after incubating plates at 37°C for 24 h.

## Statistical analysis

Triplicate trials for each experiment were performed. The data were expressed as the mean ± standard deviation (SD) and subjected to R statistical software (Version 3.4.1 for Mac OS X; http://www.r-project.org) for the Tukey-Kramer's multiple comparison test to determine statistical significance ($P \leq 0.05$).

## Results

### Determination of glass transition temperature ($T_g$)

As a representative result, changes in the compressive stress over rising temperature were shown by TRA for *S*. Typhimurium (Fig 2). The glass transition temperature ($T_g$) was

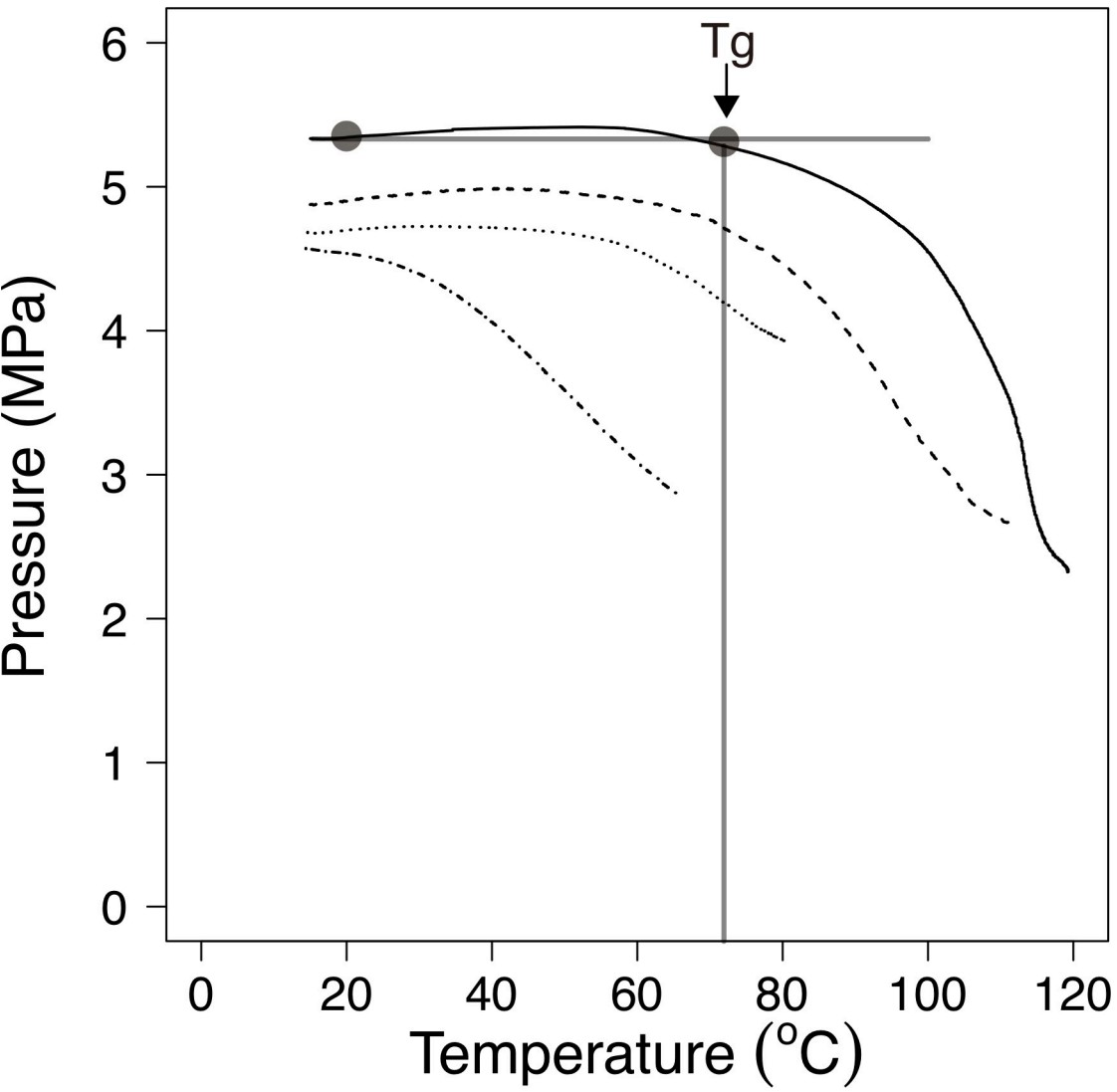

**Fig 2. Measurement of glass transition temperature ($T_g$) of *Salmonella* Typhimurium at $a_w$ = 0.43 (solid line), 0.57 (dashed line), 0.75 (dotted line), and 0.87 (dot dashed line) using thermal rheological analysis.** The onset temperature of the pressure reduction was regarded as the $T_g$ of the sample as shown in the figure.

determined as the onset temperature of the pressure reduction in the obtained compressive stress vs. temperature curve. Clear softening behaviour was observed during the temperature rising operation at $a_w$ 0.43, 0.57, 0.75, and 0.87. Similar results of the TRA analysis were observed in the other *S. enteriaca* serovars examined in this study. Accordingly, the results showed the possibility of determining bacterial cells $T_g$ by using TRA.

### Glass transition temperature ($T_g$) under each water activity condition

The $T_g$ for each *Salmonella* serovar decreased with increasing $a_w$ of the bacterial cells (Table 1). For example, for *S. enterica* Typhimurium $T_g$ was 73.7˚C, 58.6˚C, 48.4˚C, and 23.2˚C at 0.43, 0.57, 0.75, and 0.87 $a_w$, respectively. Among the five strains, *S. enterica* Chester exhibited the highest glass transition temperature at low $a_w$, whereas *S. enterica* Stanley exhibited the highest glass transition temperature at high $a_w$. Compared to other *Salmonella* serovars, *S.* Stanley showed less change in glass transition temperature with increasing $a_w$.

**Table 1. Relationship between water activity ($a_w$) and observed glass transition temperature ($T_g$) of *Salmonella enterica* serovar Typhimurium, *S.* Chester, *S.* Oranienburg, *S.* Stanley, and *S.* Enteritidis.**

| *S. enterica* serovars | Glass transition temperature ($T_g$,°C) | | | |
|---|---|---|---|---|
| | 0.43 $a_w$ | 0.57 $a_w$ | 0.75 $a_w$ | 0.87 $a_w$ |
| *S.* Typhimurium | 81.63 ± 11.28[aA#] | 70.39 ± 3.26[abA] | 58.46 ± 2.23[abB] | 35.16 ± 2.92[bC] |
| *S.* Chester | 83.29 ± 2.04[aA] | 75.62 ± 2.64[aA] | 62.17 ± 1.37[bB] | 43.50 ± 2.35[cC] |
| *S.* Oranienburg | 81.26± 6.46[aA] | 73.14 ± 1.95[abB] | 63.39 ± 4.98[bC] | 43.00 ± 1.33[cD] |
| *S.* Stanley | 80.07 ± 2.96[aA] | 65.94 ± 3.64[abB] | 62.22 ± 3.31[abB] | 57.46 ± 8.06[bB] |
| *S.* Enteritidis | 77.10 ± 1.78[aA] | 51.25 ± 2.89[bB] | 45.37 ± 2.83[cBC] | 40.69 ± 2.76[cC] |

[#]Different lowercase letters in each column ($a_w$ level) represent statistically significant differences ($P < 0.05$) among the five *S. enterica* serovars. Likewise, the different uppercase letters in each row (*S. enterica* serovar) represent statistical differences ($P < 0.05$) among four different $a_w$ levels.

## Thermal inactivation under each water activity condition

The bacterial survival ratio increased with decreasing $a_w$ levels at 60°C for 10 min (Fig 3), which would reflect rising $T_g$. The number of surviving *Salmonella* strains after thermal inactivation at 60°C for 10 min decreased by ca. 1–5 log cycles at 0.87 $a_w$. There is apparent difference in thermal tolerance among the five *S. enterica* serovars at 0.87 $a_w$. In contrast, under low-$a_w$ conditions (e.g., 0.43 $a_w$) the decrease in bacteria numbers after inactivation was ca. 1–2 log cycles in all the *S. enterica* serovars. *S. enterica* Stanley exhibited the smallest difference in survival ratio among $a_w$ levels and also higher heat resistance than the other strains. Since *S. enterica* Stanley has a $T_g > 60$°C across all of the $a_w$ as shown in the Table 1, the 60°C treatment was not sufficient to affect the bacterial inactivation. There are few differences in bacterial inactivation effect between 0.43 $a_w$ (Fig 3A) and 0.75 $a_w$ (Fig 3B) for all the *S. enterica* serovars except for *S.* Enteritidis. The apparent differences were observed in bacterial inactivation effect between 0.87 $a_w$ and the other two lower $a_w$, although there was a difference in the inactivation effect among the five serovars. This result would be linked with the relationship between aw and $T_g$ as shown in the Table 1. Namely, the significant decrease in the $T_g$ from 0.75 $a_w$ to 0.87 $a_w$ (Table 1) would result in the increase in the thermal inactivation effect (Fig 3C). There is a possibility that the bactericidal effect of pathogenic bacteria decreases in a low-$a_w$ environment.

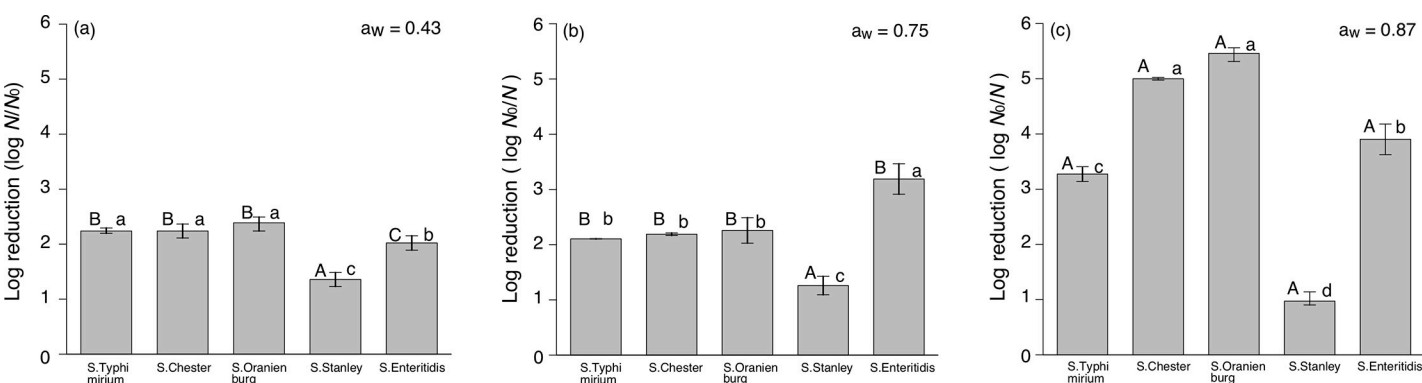

**Fig 3.** Comparison of thermal inactivation effect (log reductions) of *Salmonella enterica* Typhimurium, *S.* Chester, *S.* Oranienburg, *S.* Stanley, and *S.* Enteritidis heated at 60°C for 10 min under 0.43 (a), 0.75 (b), and 0.87 (c) $a_w$. Initial cell numbers ($N_0$) right before heat treatment of dried bacterial cells are ranged 5–7 log CFU/mL depending on serovar and $a_w$. Error bars represent standard error of the mean (n = 3). Different lowercase letters among the five *S. enterica* serovars at the same $a_w$ level represent statistically significant differences (P < 0.05). Likewise, different uppercase letters among different $a_w$ levels for each *S. enterica* serovar represent statistically significant differences (P < 0.05).

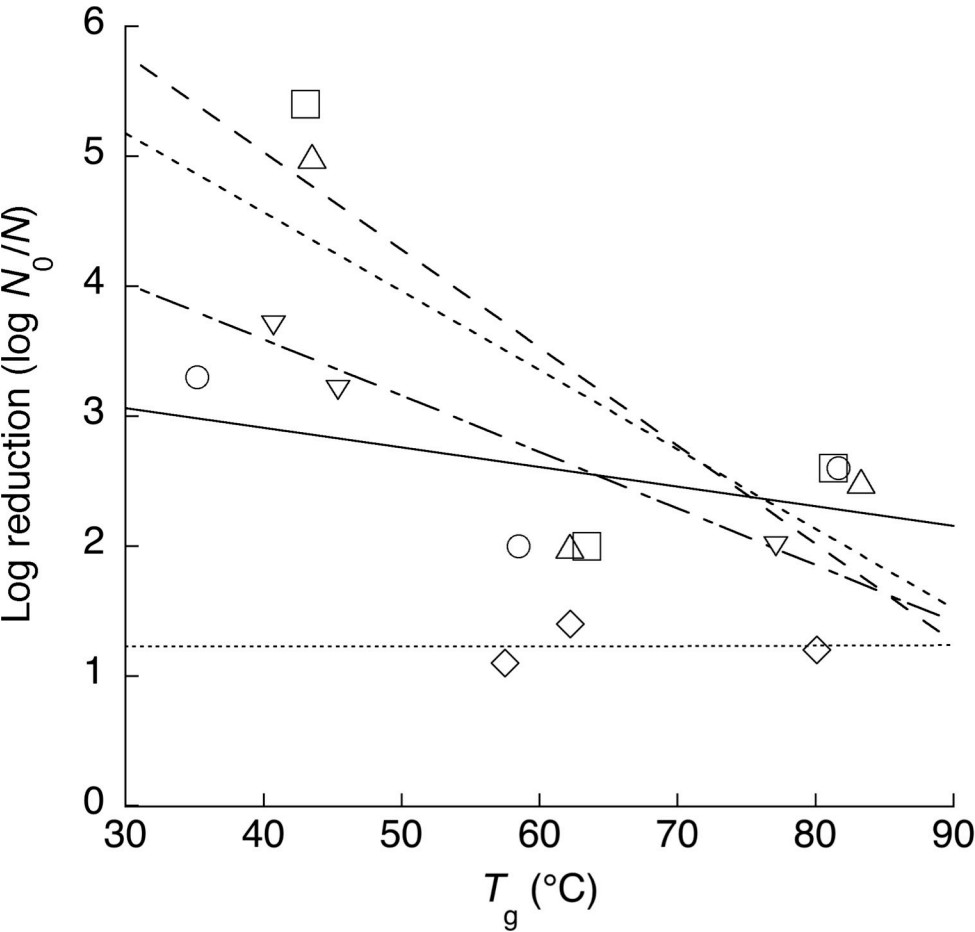

**Fig 4.** Relationship between glass transition temperature ($T_g$) and inactivation effect (log reductions) of *Salmonella* Typhimurium (○, solid line), *S.* Chester (△, dashed line), *S.* Oranienburg (□, dotted line), *S.* Stanley (◇, fine dotted line), and *S.* Enteritidis (▽, dot dashed line).

To determine the relationship between $T_g$ and heat resistance (log reduction) of *S. enterica* serovars, we illustrate both the relationship as shown in Fig 4. There seems to be $T_g$ dependency on heat resistance, which means bacterial inactivation effect increases with decrease in $T_g$ for the tested *S. enterica* serovars except for *S.* Stanly. The correlation coefficient of *S.* Typhimurium, *S.* Chester, *S.* Oranienburg, *S.* Stanley, and *S.* Enteritidis is -0.54, -0.75, -0.80, 0.01, and -0.99, respectively. The result would also support that the $T_g$ plays an important role in heat resistance of *S. enterica* cells in dry conditions.

## Discussion

Previous studies have reported that $a_w$ has some influence on bacterial survival. Long-term survival of bacteria was demonstrated in low-$a_w$ conditions ($\leq 0.87$ $a_w$), whereas bacterial death was promoted in high $a_w$ conditions [17,23,24]. The relationship between bacterial cell $T_g$ and $a_w$ is likely a major factor influencing differences in survival across a range of $a_w$ levels. The $T_g$ of *Salmonella* tested at $a_w \leq 0.87$ was 30°C or higher in the present study. These cells would be in a glass state under room temperature conditions, because room temperature (normally 20–22°C) is lower than those of $T_g$ (30°C). In a glass state, the molecular movement in bacterial cells is almost stopped and, thus, unlikely to be affected by external environments. It

is inferred that bacteria acquire desiccation tolerance by glass transition accompanying a decrease in $a_w$. Under high-$a_w$ conditions, it is presumed that $T_g$ would be considerably low, glass transition would not occur, and the rubber state would be maintained. Since molecular movement is not limited in the rubber state, bacterial cells would not acquire desiccation tolerance. We assume that this state change is a key factor in the survival differences among bacteria. We preliminary examined thermal inactivation effect on some *S. enterica* serovars, and we confirmed apparently higher inactivation effect of 6–7 log cycle reductions in aw 0.99 than those of lower $a_w$ levels on the same heat treatment (data not shown). Furthermore, since a negative correlation was demonstrated between $T_g$ and $a_w$ in *Salmonella* cells (Fig 4), there is a possibility that the bacteria will exhibit stronger desiccation tolerance as the $a_w$ decreases. In addition, since $T_g$ varied among bacterial species, the difference in desiccation tolerance will depend on the $T_g$.

This study also showed that the thermal inactivation effect decreased in low-$a_w$ conditions (Fig 3). It has been reported that the thermal inactivation effect of low-$a_w$ food [34–37]. The difference in thermal inactivation among different $a_w$ levels is likely involved in the changing physical state properties of bacterial cells as well as in their survival differences under dry conditions. As described above, bacterial cells in a low-$a_w$ environment will be in a glass state and exhibit high tolerance to environmental stresses such as heat, pressure, and desiccation. For example, extreme environment microorganisms, such as tardigrades and sleeping chironomids that utilize cryptobiosis, are also resistant to high temperature, high pressure, as well as dry environments [25–27]. Bacterial cells would vitrify, similar to extreme environmental organisms acquiring environmental stress tolerance. Therefore, we attribute the reduced thermal bacterial inactivation in low-$a_w$ conditions to a change in physical properties due to glass transition of bacterial cells.

The differences in bacterial survival (Fig 3) could be attributed to the difference in $T_g$ of each bacterium. *S. enterica* Stanley was shown to have a higher $T_g$ than the other *Salmonella* strains at high $a_w$ (Table 1), which might mean differences in the ability to maintain the glass state. In other words, *S.* Stanley would have stronger heat-tolerance than the other *Salmonella* strains. In a previous study, *S.* Stanley was reported to have a higher long-term survival ratio in dry conditions and *S.* Typhimurium showed the lowest $T_g$ at high $a_w$, which was associated with a low survival rate [17]. The difference in $T_g$ among bacterial species/serovars would attribute to innate (genetically) or acquired characteristics of each bacterial species/serovars. In particular, acquired characteristics might be due to habituation to various harsh conditions during survival process. Based on all of these findings, we believe that bacterial acquisition of environmental tolerance and the glass transition phenomenon are closely related. Although the mechanism by which $a_w$ exerts its influence on bacterial survival under desiccation and thermal conditions has not been clearly elucidated, the present study demonstrates that the glass transition phenomenon of bacterial cells may play an important role in stressful environments. Furthermore, we have successfully demonstrated that glass transition temperature will have an influence on the strength of desiccation and thermal tolerance of bacteria. To elucidate the exact reason for the difference in $T_g$ among bacterial species/serovars, further genetical and/or bacteriological investigation will be needed in the future.

In this study, we aimed to elucidate the role of the glass transition phenomenon in pathogenic bacteria obtaining tolerance under low-$a_w$ conditions. Experimental results not only confirmed the glass transition phenomenon of bacterial cells by thermal rheological analysis but also showed a clear correlation between $T_g$ and $a_w$. In addition, it was confirmed that the heat sterilization effect was reduced by vitrification of bacterial cells. These results revealed that the glass transition phenomenon of bacterial cells is a major factor in the acquisition of bacterial stress tolerance.

## Acknowledgments

We would like to thank Editage for English language editing (https://www.editage.jp).

## Author Contributions

**Conceptualization:** Kiyoshi Kawai, Shigenobu Koseki.

**Data curation:** Kyeongmin Lee, Masaki Shoda, Shigenobu Koseki.

**Formal analysis:** Kyeongmin Lee, Masaki Shoda.

**Funding acquisition:** Kiyoshi Kawai, Shigenobu Koseki.

**Investigation:** Kyeongmin Lee, Masaki Shoda, Shigenobu Koseki.

**Methodology:** Kyeongmin Lee, Shigenobu Koseki.

**Project administration:** Kiyoshi Kawai, Shigenobu Koseki.

**Resources:** Shigenobu Koseki.

**Supervision:** Shigenobu Koseki.

**Writing – original draft:** Kyeongmin Lee, Masaki Shoda, Shigenobu Koseki.

**Writing – review & editing:** Kyeongmin Lee, Masaki Shoda, Kiyoshi Kawai, Shigenobu Koseki.

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
