## [Decision Letter · Decision Letter 0]

22 Apr 2020

PONE-D-20-08834

Relationship between glass transition temperature and desiccation/heat tolerance in Salmonella enterica

PLOS ONE

Dear Prof. Koseki,

Thank you for submitting your manuscript to PLOS ONE. After careful consideration, we feel that it has merit but does not fully meet PLOS ONE’s publication criteria as it currently stands. Therefore, we invite you to submit a revised version of the manuscript that addresses the points raised during the review process.

Two reviewers have commented on your work. You will see that one raised a couple of concerns that should be dealt with before that improved version can be accepted. Please, address these concerns and resubmit a revised manuscript.

We would appreciate receiving your revised manuscript by Jun 06 2020 11:59PM. To enhance the reproducibility of your results, we recommend that if applicable you deposit your laboratory protocols in protocols.io, where a protocol can be assigned its own identifier (DOI) such that it can be cited independently in the future. For instructions see: http://journals.plos.org/plosone/s/submission-guidelines#loc-laboratory-protocols

We look forward to receiving your revised manuscript.

Kind regards,

Anderson de Souza Sant'Ana, PhD

Academic Editor

PLOS ONE

Journal Requirements:

'The funders had no role in study design, data collection and analysis, decision to publish, or preparation of the manuscript.'

Please clarify the sources of funding (financial or material support) for your study. List the grants or organizations that supported your study, including funding received from your institution.State what role the funders took in the study. If the funders had no role in your study, please state: “The funders had no role in study design, data collection and analysis, decision to publish, or preparation of the manuscript.”If any authors received a salary from any of your funders, please state which authors and which funders.If you did not receive any funding for this study, please state: “The authors received no specific funding for this work.”Please include your amended statements within your cover letter; we will change the online submission form on your behalf.

'This work was supported by the Japan Society for the Promotion of Science (JSPS) KAKENHI (Grant JP 18H02148) granted to SK.'

Reviewers' comments:

Reviewer's Responses to Questions

**Comments to the Author**

1. Is the manuscript technically sound, and do the data support the conclusions?

Reviewer #1: No

Reviewer #2: Yes

2. Has the statistical analysis been performed appropriately and rigorously? 

Reviewer #1: No

Reviewer #2: I Don't Know

3. Have the authors made all data underlying the findings in their manuscript fully available?

Reviewer #1: Yes

Reviewer #2: Yes

4. Is the manuscript presented in an intelligible fashion and written in standard English?

Reviewer #1: Yes

Reviewer #2: Yes

5. Review Comments to the Author

Reviewer #1: PONE-D-20-08834

Relationship between glass transition temperature and desiccation/heat tolerance in Salmonella enterica

Summary: In this study, Lee et al investigate the relationship between glass transition temperature (Tg) and desiccation and heat resistance in Salmonella. The main claim of the paper is that the heat and desiccation resistance of Salmonella can be attributed to the inherent Tg of the strains. This is an interesting hypothesis and has relevance to the field however, there are a number of major issues (see list below) that require attention before this manuscript can be published.

1. Conclusions not being supported by data

- The authors assert that the observed differences in heat resistance are caused by differences in Tg however the data presented do not support this claim. The relationship between Tg and aw and aw and heat resistance is shown, but the relationship between Tg and heat resistance is not. There could be many other factors unrelated to Tg that can explain the heat resistance results.

- Contrary to line 186, Salmonella ser. Stanley has an Tg less than 60 in three of the four aw conditions. It also has the highest heat tolerance. This is contradicts the hypothesis and the main assertion of the paper (lines 234-235)– please discuss.

2. Technical issues

- The aw of the stored cells should be confirmed prior to measurement of the Tg

- Figure 3 could be better presented as a table so all of the values can be compared. Also it would be helpful if there was an analysis of significant differences in Tg at different aw

- Figure 4: significant differences of the heat resistance at different aw for each strain should be determined to allow for easier comparison

- It is not clear when the initial cell numbers for the heat tolerance assay wee determined. From the way the paper reads, it seems as if the cells were at 10E7 before the freeze-drying/storage procedure. If this is the case, the cell numbers/viability need to be confirmed prior to the heat tolerance assay or else the results obtained could just be a reflection of cell viability after freeze drying

- A control of Salmonella at typical (> 0.99) aw would be helpful to have as a comparator for the Tg and heat tolerance results at low aw

- A rationale for why the strains/serovars were chosen for this study is not given

3. Other issues

- Background: As someone who does not specialize in rheological measurements, I found that the information provided in the Introduction and Methods was insufficient and did not allow for the appropriate review of the methods used to determine Tg.

- There was little description of these measurements being done in a microbiological context and their applicability to bacterial cells. A discussion of the physiological state of vitrified cells would be helpful – are these cells considered to be analogous to VBNC (viable but not culturable cells)?

- Line 211, Room temperature is generally 20 – 22 degrees C and not 30. This has implications concerning the assertion that Salmonella is in a glass state at room temperature

Reviewer #2: General comment:

The manuscript presents a study investigating the relationship or effect of glass transition temperature (Tg) under different water activity (aw) conditions using alternative thermal rheological analysis (TRA). The central aim was to test the survival of Salmonella strain in various conditions. This was based on a hypothesis that bacterial cells are vitrified in low-aw environments based on the physicochemical properties of solid particles.

Specific comment:

Topic should change to: Relationship between glass transition temperature, desiccation and heat tolerance in Salmonella enterica

Abstract did not present specific aspects of the result found. it only mention the result without details. For instance, the abstract must contain a summary of the result (temperature rising operation at aw 0.43, 0.57, 0.75, and 0.87). and the result for Tg (For example, for S. enterica Typhimurium Tg was 73.7 ºC, 58.6 ºC, 48.4 ºC, and 23.2 166 ºC at 0.43, 0.57, 0.75, and 0.87 aw, respectively).

Discussion:

Line 220-221: The current study observed a variation in Tg among bacterial species. There is need to offer a scientific explanation on why various species of bacteria reacted differently to Tg. Little explanation was offered in line 236, this is sufficient and convincing that all Salmonella strains have been shown to survive in specific conditions.

Line 223-224: In fact, it has been reported that the thermal inactivation effect of low-aw food also [34-37].

Should be corrected to

It has been reported that the thermal inactivation effect of low-aw food [34-37].

6. PLOS authors have the option to publish the peer review history of their article (what does this mean?). If published, this will include your full peer review and any attached files.

Reviewer #1: No

Reviewer #2: No

---

## [Author Response · Author response to Decision Letter 0]

1 May 2020

May 1, 2020

PONE-D-20-08834

Title: Relationship between glass transition temperature and desiccation/heat tolerance in Salmonella enterica

Dear Dr. Sant'Ana, 

Academic Editor, PLOS ONE,

Thank you for your email on April 22, 2020 regarding our manuscript PONE-D-20-08834. We greatly appreciate the reviewers’ comments and suggestions that have helped us improve our manuscript. We have revised the manuscript corresponding to the reviewers’ comments. Our point-by-point responses to the reviewers’ comments are appeared below.

Reviewer #1

Summary: In this study, Lee et al investigate the relationship between glass transition temperature (Tg) and desiccation and heat resistance in Salmonella. The main claim of the paper is that the heat and desiccation resistance of Salmonella can be attributed to the inherent Tg of the strains. This is an interesting hypothesis and has relevance to the field however, there are a number of major issues (see list below) that require attention before this manuscript can be published.

1. Conclusions not being supported by data

- The authors assert that the observed differences in heat resistance are caused by differences in Tg however the data presented do not support this claim. The relationship between Tg and aw and aw and heat resistance is shown, but the relationship between Tg and heat resistance is not. There could be many other factors unrelated to Tg that can explain the heat resistance results.

RESPONSE 

Thank you for your critical comment. In fact, there was no direct relationship between Tg and heat resistance in the original manuscript. Based on the relationship between Tg and aw (Fig. 3 in the original manuscript), and the relationship between aw and heat resistance (Fig. 4 in the original manuscript), we considered that the relationship between Tg and heat resistance would be illustrated indirectly. However, as the Reviewer#1 concerns, it would be unclear the relationship between Tg and heat resistance. Accordingly, we have added one more figure that illustrates directly the relationship between Tg and heat resistance (log reduction) as a new Figure 4 in the revised manuscript as shown below. The result shows the trend that bacterial inactivation effect increases with decrease in Tg, although there is variation among serovars. 

Fig. 4. Relationship between glass transition temperature (Tg) and inactivation effect (log reductions) of Salmonella Typhimurium (○, solid line), S. Chester (△, dashed line), S. Oranienburg (□, dotted line), S. Stanley (◇, fine dotted line), and S. Enteritidis (▽, dot dashed line). 

“To determine the relationship between Tg and heat resistance (log reduction) of S. enterica serovars, we illustrate both the relationship as shown in Fig. 4. There seems to be Tg dependency on heat resistance, which means bacterial inactivation effect increases with decrease in Tg for the tested S. enterica serovars except for S. Stanly. The correlation coefficient of S. Typhimurium, S. Chester, S. Oranienburg, S. Stanley, and S. Enteritidis is -0.54, -0.75, -0.80, 0.01, and -0.99, respectively. The result would also support that the Tg plays an important role in heat resistance of S. enterica cells in dry conditions.” (Please see Lines 212-218 in the revised manuscript.)

- Contrary to line 186, Salmonella ser. Stanley has an Tg less than 60 in three of the four aw conditions. It also has the highest heat tolerance. This is contradicts the hypothesis and the main assertion of the paper (lines 234-235)– please discuss.

RESPONSE 

Thank the Reviewer#1 for the critical comments. As the Reviewer #1 indicated, the data presented in the original manuscript would lead to some confusions, because of the incorrect descriptions on the figure caption of the Figure 3 due to our careless mistake during revising the figure. Accordingly, we corrected the figure captions, however, we re-arranged the results based on the Reviewer #1’s another comments below regarding transforming Fig. 3 into a Table. Please see the Response to the comment 2.2 in below. 

According to the correction, S. Stanley has higher glass transition temperature at aw 0.87, unlike other Salmonella strains. As shown in Fig. 4 in the original manuscript (the figure will refer to Fig. 3 in the revised manuscript), when heat treatment of 60 ºC was applied, the Salmonella strains other than S. Stanley showed a higher log reduction at aw 0.87 than those of other aw levels. The difference in the bacterial inactivation effect between aw levels would be related to the Tg. In contrast, S. Stanley did not change log reduction of bacteria depending on increasing water activity. This is considered that S. Stanley has a high glass transition temperature even at high water activity, and as a result, it is high possibility to be in a glass state at a temperature of 60 ºC in aw 0.87, and thus has high heat resistance.

2. Technical issues

2.1- The aw of the stored cells should be confirmed prior to measurement of the Tg

RESPONSE 

The water activity and temperature in the air-tight container were always checked with a temperature recorder, and the water activity of the bacteria was confirmed with a water activity meter. Therefore, we have added the following text.

“ The water activity and temperature in the air-tight container were continuously checked using thermo recorder (TR-72wf, T and D, Nagano, Japan). And the water activity of the bacteria was confirmed by a water activity meter (Aqualab 4TE, Decagon Devices, Washington, USA).” (Please see Lines 117-120 in the revised manuscript.)

2.2 Figure 3 could be better presented as a table so all of the values can be compared. Also it would be helpful if there was an analysis of significant differences in Tg at different aw

RESPONSE 

Thank you for your suggestion. We have changed Fig. 3 into Table 1 as fallows to add the information regarding the significant difference between aw levels in each Salmonella serotype as well as the difference between Salmonella serotypes in each aw as shown in the original figure. 

Please see the Table 1 as shown in below in the revised manuscript and Lines 178, and 185 – 191. 

Table 1. Relationship between water activity (aw) and observed glass transition temperature (Tg) of Salmonella enterica serovar Typhimurium, S. Chester, S. Oranienburg, S. Stanley, and S. Enteritidis.

2.3 Figure 4: significant differences of the heat resistance at different aw for each strain should be determined to allow for easier comparison

RESPONSE 

We appreciate the Reviewer #1’s constructive suggestion. We have added the information regarding the significant difference between aw levels in each Salmonella serotype as different uppercase letters. Please see the revised Figure 4 referred as Figure 3 in the revised manuscript and the caption of the Figure 3 in Lines 220 – 227 in the revised manuscript as follows:

“Fig. 3. Comparison of thermal inactivation effect (log reductions) of Salmonella enterica Typhimurium, S. Chester, S. Oranienburg, S. Stanley, and S. Enteritidis heated at 60 ºC for 10 min under 0.43 (a), 0.75 (b), and 0.87 (c) aw. Initial cell numbers (N0) right before heat treatment of dried bacterial cells are ranged 5 – 7 log CFU/mL depending on serovar and aw. Error bars represent standard error of the mean (n = 3). Different lowercase letters among the five S. enterica serovars at the same aw level represent statistically significant differences (P < 0.05). Likewise, different uppercase letters among different aw levels for each S. enterica serovar represent statistically significant differences (P < 0.05). ”

2.4 It is not clear when the initial cell numbers for the heat tolerance assay were determined. From the way the paper reads, it seems as if the cells were at 10E7 before the freeze-drying/storage procedure. If this is the case, the cell numbers/viability need to be confirmed prior to the heat tolerance assay or else the results obtained could just be a reflection of cell viability after freeze drying

RESPONSE 

Thank you for the critical indication. As the Reviewer #1 mentioned, the description on the initial cell number in the original manuscript was unclear on the timing of determination of the initial cell numbers. The initial cell numbers we mentioned here were determined for the dried and aw-adjusted bacterial cells just before heat treatment. Depending on the S. enterica serovars, the viable cell numbers showed difference after adjustment of aw. That is why we evaluated the heat tolerance as log-reduction to minimize the difference of the initial viable cell numbers. 

To clarify the meaning of initial cell numbers for the heat treatment experiment, we have modified the explanation of the method section.

“In the same manner, dried bacterial cells (ca. 50 mg) were taken and determined the viable cell number as an initial condition.”

Please see Lines 143 – 144 in the revised manuscript. 

2.5 A control of Salmonella at typical (> 0.99) aw would be helpful to have as a comparator for the Tg and heat tolerance results at low aw

RESPONSE 

In fact, glass transition phenomenon is observed only lower aw conditions such as < 0.93. We could not determine the Tg for the tested Salmonella > 0.93 aw. Therefore, we focus on low water activity to explain mechanism of these strain that tolerate in low water activity condition. 

Preliminary, some S. enterica serovars were examined at the same heat treatment in the aw 0.99. We confirmed apparently higher inactivation effect of 6-7 log cycle reductions in aw 0.99 than those of lower aw levels on the same heat treatment. Thus, we added the descriptions regarding inactivation effect at typical aw level as follows:

“We preliminary examined thermal inactivation effect on some S. enterica serovars, and we confirmed apparently higher inactivation effect of 6-7 log cycle reductions in aw 0.99 than those of lower aw levels on the same heat treatment (data not shown).”

Please see Lines 249-252 in the revised manuscript. 

2.6 A rationale for why the strains/serovars were chosen for this study is not given

RESPONSE 

Thank you for your suggestion. We selected these strains of Salmonella known to exist in dry foods and related to outbreaks. We have added it accordingly as follows:

“Especially, various Salmonella serotypes were found in dry foods associated with outbreaks [1-12].” (Please see Lines 42-43)

“Finally, we aimed to resolve the relationship between the state change of several Salmonella serotypes that is known to be present in low water activity foods due to glass transition and the changes in thermal resistance in a desiccation environment.” (Please see Lines 87-90)

3. Other issues

3.1 Background: As someone who does not specialize in rheological measurements, I found that the information provided in the Introduction and Methods was insufficient and did not allow for the appropriate review of the methods used to determine Tg.

RESPONSE 

We added more detail descriptions as follows:

“Previous studies used by TRA investigated the effect of water content on the Tg of cookies (Kawai et al., 2014; Sogabe et al., 2018), hazelnuts (Ebara et al., 2018), and deep-fried food (Jothi et al., 2018). To conduct the measurements, a sample is compressed at a temperature below Tg, and heated above Tg with compression. Then, the Tg of the sample can be determined as a force drop induced by the glass transition. This is a useful method to apply to amorphous powders. ”

Please see Lines 81-85 in the revised manuscript. 

3.2 There was little description of these measurements being done in a microbiological context and their applicability to bacterial cells. A discussion of the physiological state of vitrified cells would be helpful

RESPONSE 

To the best of our knowledge, the present study would be a first investigation based on the hypothesis that bacterial cells vitrify depending on environmental aw level. Thus, there is few related literature and we could not discuss the physiological state of the cell beyond our speculation. Therefore, as only the related information, we discussed on tardigrades and sleeping chironomids that utilize cryptobiosis, are also resistant to high temperature, high pressure, as well as dry environments (Lines 262-264 in the revised manuscript). We could not discuss in detail, however, we do believe that the present study would be a trigger of further investigation of the vitrification of bacterial cells. 

 3.3 are these cells considered to be analogous to VBNC (viable but not culturable cells)?

RESPONSE 

It is still unclear the relationship between glass transition and VBNC. However, we suppose that glass transition might be a part of VBNC state. When it becomes glassy state, the activity of the cells decreases, so it has something in common with the VBNC bacteria. However, it is necessary to study further because other factors besides water activity must be considered in order to judge these bacteria as VBNC bacteria.

3.4 Line 211, Room temperature is generally 20 – 22 degrees C and not 30. This has implications concerning the assertion that Salmonella is in a glass state at room temperature

RESPONSE 

As a result of measuring Tg in this study, the Tg of Salmonella serovars showed more than 30 ºC when the water activity was � 0.87. We consider that Salmonella would be glassy state in the room temperature (20-22 ºC) because room temperature was lower than Tg (>30 ºC). Thus, we have modified the sentence as follows:

“These cells would be in a glass state under room temperature conditions, because room temperature (normally 20 -22 ºC ) is lower than those of Tg (30 ºC). ” (Please see Lines 241 – 242 in the revised manuscript.)

Thank you again for the Reviewer #1’s helpful comments and suggestions.

 

Reviewer #2

Reviewer #2: General comment:

The manuscript presents a study investigating the relationship or effect of glass transition temperature (Tg) under different water activity (aw) conditions using alternative thermal rheological analysis (TRA). The central aim was to test the survival of Salmonella strain in various conditions. This was based on a hypothesis that bacterial cells are vitrified in low-aw environments based on the physicochemical properties of solid particles.

Specific comment:

1. Topic should change to: Relationship between glass transition temperature, desiccation and heat tolerance in Salmonella enterica

RESPONSE

We have replaced the title accordingly as follows:

“Relationship between glass transition temperature, and desiccation and heat tolerance in Salmonella enterica” (Please see Lines 1-2)

2. Abstract did not present specific aspects of the result found. it only mention the result without details. For instance, the abstract must contain a summary of the result (temperature rising operation at aw 0.43, 0.57, 0.75, and 0.87). and the result for Tg (For example, for S. enterica Typhimurium Tg was 73.7 ºC, 58.6 ºC, 48.4 ºC, and 23.2 166 ºC at 0.43, 0.57, 0.75, and 0.87 aw, respectively).

RESPONSE

Thank you for your suggestion. We have added next sentence as follows:

“For example, while the Tg of five S. enterica serovars was determined as 35.16ºC to 57.46ºC at 0.87 aw, the Tg of all the five serovars increased to 77.10ºC to 83.30ºC at 0.43 aw. ” (Please see Lines 28-30 in the revised manuscript.)

Discussion:

3. Line 220-221: The current study observed a variation in Tg among bacterial species. There is need to offer a scientific explanation on why various species of bacteria reacted differently to Tg. Little explanation was offered in line 236, this is sufficient and convincing that all Salmonella strains have been shown to survive in specific conditions.

RESPONSE

Thank you for your critical comments. In fact, we do not confirm any exact scientific reason/evidence for the difference among bacterial species/serovars. However, there might be two possible reasons. One of them would be due to properties that are genetically unique to the bacterial species/serovars. Another one would be due to properties that are acquired during survival in various harsh environments. In any case, the reason/evidence for the difference in Tg among species/serovars should be investigated in the future study. 

“The difference in Tg among bacterial species/serovars would attribute to innate (genetically) or acquired characteristics of each bacterial species/serovars. In particular, acquired characteristics might be due to habituation to various harsh conditions during survival process. To elucidate the exact reason for the difference in Tg among bacterial species/serovars, further genetical and/or bacteriological investigation will be needed in the future. “

Please see Lines 274 – 277 and 284 – 286 in the revised manuscript. 

4. Line 223-224: In fact, it has been reported that the thermal inactivation effect of low-aw food also [34-37].

Should be corrected to

It has been reported that the thermal inactivation effect of low-aw food [34-37].

RESPONSE

We have corrected accordingly as follows:

“It has been reported that the thermal inactivation effect of low-aw food [34-37]. ” (Please see Lines 257 – 258 in the revised manuscript.

Thank you again for the Reviewer #2’s helpful comments and suggestions.

---

## [Decision Letter · Decision Letter 1]

11 May 2020

Relationship between glass transition temperature, and desiccation and heat tolerance in Salmonella enterica

PONE-D-20-08834R1

Dear Dr. Koseki,

We are pleased to inform you that your manuscript has been judged scientifically suitable for publication and will be formally accepted for publication once it complies with all outstanding technical requirements.

With kind regards,

Anderson de Souza Sant'Ana, PhD

Academic Editor

PLOS ONE

Additional Editor Comments (optional):

Reviewers' comments:

Reviewer's Responses to Questions

**Comments to the Author**

1. If the authors have adequately addressed your comments raised in a previous round of review and you feel that this manuscript is now acceptable for publication, you may indicate that here to bypass the “Comments to the Author” section, enter your conflict of interest statement in the “Confidential to Editor” section, and submit your "Accept" recommendation.

Reviewer #1: All comments have been addressed

Reviewer #2: All comments have been addressed

2. Is the manuscript technically sound, and do the data support the conclusions?

Reviewer #1: Yes

Reviewer #2: Partly

3. Has the statistical analysis been performed appropriately and rigorously? 

Reviewer #1: Yes

Reviewer #2: Yes

4. Have the authors made all data underlying the findings in their manuscript fully available?

Reviewer #1: Yes

Reviewer #2: No

5. Is the manuscript presented in an intelligible fashion and written in standard English?

Reviewer #1: Yes

Reviewer #2: Yes

6. Review Comments to the Author

Reviewer #1: All of my comments have been addressed. The manuscript reads much better and I recommend its publication.

Reviewer #2: (No Response)

7. PLOS authors have the option to publish the peer review history of their article (what does this mean?). If published, this will include your full peer review and any attached files.

Reviewer #1: No

Reviewer #2: Yes: Dr Ishmael Festus Jaja

---

## [Editor Report · Acceptance letter]

14 May 2020

PONE-D-20-08834R1 

Relationship between glass transition temperature, and desiccation and heat tolerance in Salmonella enterica 

Dear Dr. Koseki:

I am pleased to inform you that your manuscript has been deemed suitable for publication in PLOS ONE. Congratulations! Your manuscript is now with our production department. 

With kind regards,

on behalf of

Professor Anderson de Souza Sant'Ana 

Academic Editor

PLOS ONE